# Information Recall in Pre-Operative Consultation for Glioma Surgery Using Actual Size Three-Dimensional Models

**DOI:** 10.3390/jcm9113660

**Published:** 2020-11-13

**Authors:** Sümeyye Sezer, Vitoria Piai, Roy P.C. Kessels, Mark ter Laan

**Affiliations:** 1Department of Neurosurgery, Radboud University Medical Center, 6525 GA Nijmegen, The Netherlands; Sumeyye.Sezer@radboudumc.nl; 2Department of Medical Psychology, Radboud University Medical Center, 6525 GA Nijmegen, The Netherlands; V.Piai@radboudumc.nl (V.P.); Roy.Kessels@radboudumc.nl (R.P.C.K.); 3Donders Institute for Brain, Cognition and Behaviour, Radboud University, 6525 AJ Nijmegen, The Netherlands

**Keywords:** 3D model, Augmented Reality, glioma, pre-operative consultation, patient education, information recall

## Abstract

Three-dimensional (3D) technologies are being used for patient education. For glioma, a personalized 3D model can show the patient specific tumor and eloquent areas. We aim to compare the amount of information that is understood and can be recalled after a pre-operative consult using a 3D model (physically printed or in Augmented Reality (AR)) versus two-dimensional (2D) MR images. In this explorative study, healthy individuals were eligible to participate. Sixty-one participants were enrolled and assigned to either the 2D (MRI/fMRI), 3D (physical 3D model) or AR groups. After undergoing a mock pre-operative consultation for low-grade glioma surgery, participants completed two assessments (one week apart) testing information recall using a standardized questionnaire. The 3D group obtained the highest recall scores on both assessments (Cohen’s d = 1.76 and Cohen’s d = 0.94, respectively, compared to 2D), followed by AR and 2D, respectively. Thus, real-size 3D models appear to improve information recall as compared to MR images in a pre-operative consultation for glioma cases. Future clinical studies should measure the efficacy of using real-size 3D models in actual neurosurgery patients.

## 1. Introduction

Patient education plays an essential role in patient-centered care. Effective patient education enables patients to become active participants in their health care, which is the principle of shared decision making. Their participation in decision making leads to higher patient empowerment (defined as “a process through which people gain greater control over decisions and actions affecting their health” (World Health Organization) [1]) and high-quality health care [2,3]. Patients themselves also express the desire to receive such information [4] and good communication has resulted in higher patient satisfaction and better comprehension of medical information [5]. Effective communication may also enhance recall. This would be desirable, as about 40–80% of the information patients receive from healthcare professionals is immediately forgotten [6] and retention can be as low as 13% after one month [7].

In order to achieve informed consent, the patient has to understand the given information and be able to recall it correctly. The form in which information is given to the patient affects the information recall [6,8,9]. Thus, visual tools that have the ability to improve recall, might contribute to achieving informed consent. New 3D technologies (physical 3D models, Augmented Reality (AR), Virtual Reality (VR)), are increasingly being used in medicine. The literature about the use of 3D models in patient education is limited, but it has been shown to improve several aspects of information recall [10,11,12]. 

Especially in oncological cases, it is important to visualize the tumor, as the tumor size and location determine the treatment strategy. Particularly for glioma, the use of 3D technologies may positively influence communication with the patient. That is, the main treatment option in glioma is a resection as complete as possible, without damaging healthy brain tissue. The complex relation of the tumor to healthy brain tissue is different in each patient and three-dimensional (3D) in nature. Consequently, current two-dimensional visual tools lack the ability to properly display these complex 3D relations. A personalized 3D model, however, enables visualization of the patient-specific tumor and eloquent areas. 

In a previous qualitative study about the use of 3D models in pre-operative glioma care, patients indicated that the model helped in understanding the surgical procedure and its risks [13]. Better recall was also mentioned by some patients. However, effects on recall of treatment-related information using 3D models in glioma patients have never been quantified.

In this study, we compare the amount of information recall after a mock pre-operative consult using a 3D model (physically printed or in AR) versus MR images in healthy individuals. In order to create an as optimal design as possible, our study group consists of carefully selected healthy participants with similar baseline characteristics as those of patients with a glioma. We hypothesize the percentage of information recall to be higher when using the 3D model or AR, than when using the MR images.

## 2. Materials and Methods

### 2.1. Participants

We performed an explorative study in which healthy, native Dutch speaking individuals between the age of 18 and 60 were eligible to participate. Exclusion criteria were as follows: 1. A history of neuro-oncological disease, 2. Cognitive impairment or subjective cognitive complaints due to a neurological or psychiatric disorder and 3. Brain anatomy knowledge at a high level (i.e., at the level of a neurologist, neurosurgeon, neuroscientist). To recruit individuals, a call was put out on two local platforms (Nextdoor application, Version 2.103 and the online board of the Radboud University). All participants were assigned to either the 2D (MRI/fMRI), 3D (physical 3D model) or AR group, stratified for age, sex, education and pre-existing anatomical knowledge. The research ethics committee of the Radboud University Nijmegen Medical Centre approved the study (registration number: 2019-5550).

### 2.2. Study Procedure

The experiment was set up to reflect a real-life scenario of a pre-operative consultation for low-grade glioma surgery. For this experiment, three different clinical cases of low-grade gliomas were used. Each case was linked to a specific neurosurgical resident who conducted all the consultations based on a pre-developed script (Appendix A) as to ensure a consistent content of the discussed information. Each resident conducted consultations using all three different visualization methods (2D images, 3D model and AR).

The experiment was conducted on a one-to-one basis in order to make the encounter as similar as possible to a real clinical consultation. Participants received a short scenario and were instructed to act like when they would visit the doctor themselves. Residents were instructed to act like a regular patient was visiting and to limit the consultation to a maximum of thirty minutes. After informed consent provided by the participant, consultations were audio recorded and later transcribed and analyzed for content. 

Immediately after the consultation, participants filled out Questionnaire 1 for immediate recall (Appendix B), consisting of 26 multiple choice and one open question. Questions addressed four domains of factual knowledge: 1. Tumor characteristics, 2. Diagnosis/imaging, 3. Treatment options, 4. Risks of treatment. One week after the first assessment, participants received the same questionnaire, Questionnaire 2 for 1-week follow-up. All questionnaires were sent out through Castor (Ciwit B.V., Amsterdam, The Netherlands).

### 2.3. Visualization Tools

Within the 2D group, participants were shown axial, coronal and sagittal images of the tumor and its surrounding eloquent brain areas. T1 images with and without contrast, T2/FLAIR images and functional MR images were shown. These images were obtained from actual clinical cases and were fully anonymized.

Within the 3D group, participants were shown a physical, real-size 3D model of the tumor and the eloquent brain areas as well as the (f)MRI images on the screen. To produce the 3D models, the tumor and eloquent brain areas were segmented and, together with the tractography based on diffusion-weighted images, were visualized as 3D models. These were exported as a Digital Imaging and Communications in Medicine (DICOM) image sequence. The 3D model reconstructions were made using MeVisLab (MeVis MedicalSolutions AG, Bremen, Germany). Finally, the 3D model reconstructions were exported as STL files, made ready for printing using 3DSMax 2015 (Autodesk Inc., San Rafael, CA, USA) and imported in Cura version 2.1.2 64-bit to be sent to the Ultimaker 3D printer (Ultimaker BV, Geldermalsen, The Netherlands). 

Within the AR group, an iPad Pro 11 (Apple Inc. Cupertino, CA, USA) was used to show the 3D image. The AR models were created by editing the STL files (mentioned above) in Unity (Unity Technologies, San Francisco, CA, USA). These were imported into in-house developed software, GreyMapp (© 2020, Nijmegen, the Netherlands).

An illustration of the used visual tools can be found in Figure 1.

### 2.4. Analysis

Scoring of questionnaires was done with an answer form per clinical case. Questionnaire outcomes are presented as percentages of correct answers (total sum of correct answers divided by the maximum score × 100). The maximum possible score was adjusted on a case-by-case basis after analysis of the audio transcripts when certain information had been omitted by the resident during the consultation. Differences between groups were analyzed using a repeated-measure general linear model (GLM) analysis with assessment (immediate vs 1 week) as within-subject factor and group (3D vs. 2D, AR vs. 2D and 3D vs. AR) as between-subject factor. Post hoc comparisons were performed using Fisher’s Least Significant Difference. To determine effect sizes, we computed Cohen’s d (d = 0.2 being a “small” effect, 0.5 a “medium” effect and 0.8 a “large” effect). The effect sizes for the 4 domains in the questionnaires were presented for descriptive purposes. All analyses were done using SPSS software version 26 (IBM Inc., Armonk, NY, USA).

## 3. Results

Between November 2019 and March 2020, sixty-three participants were enrolled in the study. However, as a result of the COVID-19 measures, only sixty-one participants participated. All patients completed both assessment 1 and 2, with a mean duration of 8.7 days between both assessments. There was no difference in age, level of education or pre-existing anatomical knowledge between groups (Table 1). 

A summary of the questionnaire outcomes can be found in Table 2. Figure 1 shows the outcomes of both questionnaires for each group and the spread, the latter being largest in the 2D group for both questionnaires. The questionnaire outcomes did not differ significantly across the three cases, belonging to a specific resident (immediate recall; F(2, 58) = 0.10 ; *p* = 0.909 and 1-week follow-up; F(2, 58) = 0.38 ; *p* = 0.686). 

The GLM analyses showed a main effect of assessment (F(1, 58) = 12.30; *p* = 0.001), indicating an overall worse recall after 1 week compared to immediate assessment. A main effect of group was found (F(2, 58) = 8.88; *p* < 0.0005), but no assessment × group interaction (F(2, 58) = 1.98; *p* = 0.147), indicating that the decay over time did not differ across the groups. Post hoc analyses showed that both the 3D group (mean Δ 5.66; 95% CI 2.96–8.35; *p* < 0.0005) and AR group (mean Δ 3.10; 95% CI 0.41–5.80; *p* = 0.025) obtained higher recall scores compared to the 2D group. The 3D group obtained better recall scores than the AR group, but this difference was insignificant (mean Δ 2.56; 95% CI −0.17–5.28; *p* = 0.066). 

The effects were large for the 3D model group compared to the 2D group (immediate recall: d = 1.76; 1-week follow-up: d = 0.94). Comparing the AR to the 2D group showed a large difference for immediate recall (d = 1.09), but a small difference after the 1-week follow-up (d = 0.29). Thus, the effect sizes of 3D as compared to 2D are higher than the effect of AR as compared to 2D.

Figure 2 shows effect sizes per domain for immediate recall and the 1-week follow-up recall. Per-domain analysis showed that the effect on recall ranges for each group; thus, improvement in recall is not limited to one specific domain.

## 4. Discussion

The main purpose of this study was to compare the amount of information that is recalled after a mock pre-operative consult using either a 3D model, AR models or 2D MR images. Our outcome shows that recall was highest for the 3D printed model group, followed by the AR group, with the lowest recall obtained in the 2D MRI group. 

Our findings, that both 3D methods (either physically printed or presented using augmented reality) result in superior recall than using a 2D MR images, are in line with the few previous studies about the use of 3D models in patient education. In a qualitative study, patients indicated that the use of a 3D model made it easier for them to understand their surgeon and to ask questions regarding their situation [13]. Another, quantitative study [12] described a significant larger improvement in comprehension and satisfaction when using 3D printed models (Likert score of 1.7 with SD = 0.67 to 4.7 with SD = 0.67) versus computed tomography angiography (CTA) images (1.5 with SD = 0.53 to 2.5 with SD = 0.53) for explaining cerebral aneurysm clipping. For lumbar degeneration, a prospective study on the use of 3D models and 3D reconstructions (on a 2D screen) in a pre-operative consultation for lumbar degeneration showed a better understanding in the 3D model group (15.9% compared to CT/MRI, 8.5% compared to 3D reconstruction) as well as the highest satisfaction in the 3D model group, followed by 3D reconstruction and CT/MRI (91.3%, 66.83%, and 32.8%) [10]. 

Studies on non-neurosurgical patients found similar results [11,14,15,16]. Patients verbally stated an improvement in understanding of anatomical details and surgical intervention in patients with renal malignancies [15]. Another study [16] used 3D models to explain the anatomy, disease state and treatment options in nasal sinus defects and found statistically significant improvements in patient reported (anatomical) understanding. 

Extending previous studies, we also examined subdomains of the information provided separately (Tumor characteristics, Imaging and diagnosis, Treatment options and Risks of treatment). This is relevant, as for instance the treatment options and risks are the most important aspects of a pre-operative consultation for the patient to understand, since they determine the prognosis. We found a medium to large effect size for all four domains in the 3D group, suggesting that 3D models improve recall on all information dimensions.

When comparing AR to 3D models we found that the 3D model scored consistently better on both questionnaires, but this difference was not significant. The effect size of 3D models both on immediate and 1-week-follow-up recall was higher than for the AR models (Cohen’s d = 1.76 and 0.94 (3D) vs. 1.09 and 0.29 (AR)) as compared to 2D. As shown in Figure 1, this seems especially to be caused by low performance of the AR models in the tumor characteristics domain. This finding is similar to a previous study investigating the role of 3D and AR models in renal and prostate malignancies [11,14,15]. This possibly can be explained by inexperience of patients in using AR, difficulty in translating AR images to “real life” and the added value of being able to touch and feel a physical 3D model. Considering these superior effect sizes, we recommend using physical models over AR models.

The first step in creating both 3D and AR models, is to visualize the tumor and the eloquent brain areas as a 3D model using Brainlab technology. At our institute, this step is part of the standard pre-operative workflow of the neurosurgeon and is not considered as additional work. However, the following step to translate these files into physical 3D models or AR images for the application, takes one to two hours of technician time per model. The production price of the physical 3D models depends on the used printer and materials; at our institute, materials for one model cost around 5 EUR. Considering the benefits we have found, we think these costs are acceptable. 

This experimental study is the first to compare the information recall after using a 3D model or AR models. A strength of our study is the set-up of the experiment with three different clinical cases. Another strength is the use of two assessments, that is, immediate recall and a 1-week follow-up assessment, and the use of audio recordings, which enabled us to analyze the consultations of individual participants in a reliable way. 

A limitation of our study could be the inclusion of healthy individuals rather than patients, which limits the external validity of our findings. Inclusion of healthy individuals without any cognitive impairments and who are not emotionally involved in the discussed subject might also explain the relatively high overall recall. However, we selected a study population based on similar baseline characteristics as those of patients with a glioma. Furthermore, this study provides important information for setting up a subsequent study in glioma patients. Such a study would also require detailed assessment of cognitive dysfunction (which may be the case in any brain tumor patient), and take factors such as attentional narrowing [17] and anxiety [18] into account. In healthy individuals, these factors do not play an important role and thus do not act as confounding factors. Finally, a subsequent study in glioma patients might show a different (possibly lower) percentage of information recall.

## 5. Conclusions

Real-size 3D models, either as 3D prints or as AR models, improve information recall compared to using 2D MR images in a pre-operative consultation for glioma cases. To further investigate the effectiveness of 3D models as a visual aid for patient education, clinical studies should be set up to quantitively measure patient experiences.

## Figures and Tables

**Figure 1 jcm-09-03660-f001:**
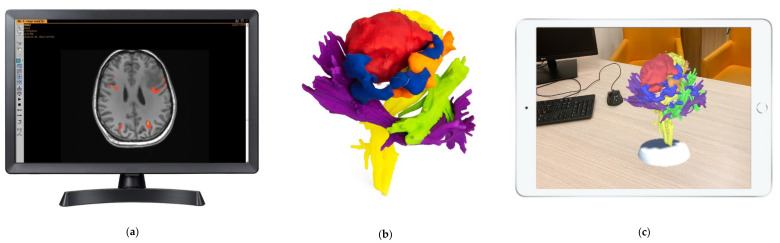
Display of all three visual tools. (**a**) Functional Magnetic Resonance Imaging (fMRI), used in the 2D group. (**b**) Printed three-dimensional (3D) model, used in the 3D group. (**c**) Augmented Reality (AR) model displayed on an iPad, used in the AR group.

**Figure 2 jcm-09-03660-f002:**
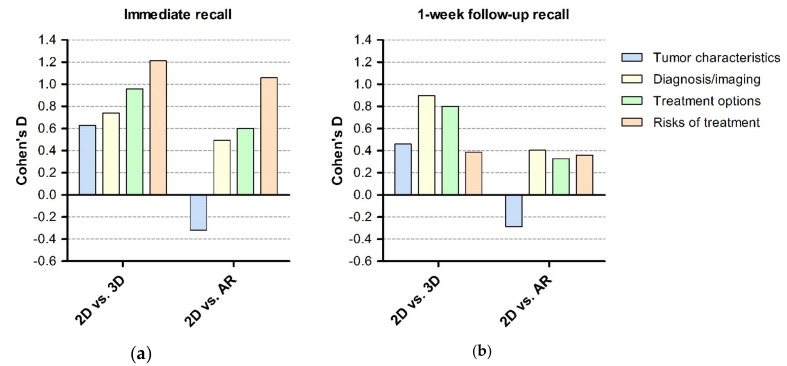
Bar graph with Cohen’s d on y-axis and comparison of groups (3D vs. 2D, AR vs. 2D and 3D vs. AR) for each of the four domains. (**a**) Immediate recall and (**b**) 1-week follow-up recall. 2D = two-dimensional group (MRI), 3D = three-dimensional group (3D model), AR = Augmented Reality group.

**Table 1 jcm-09-03660-t001:** Participant characteristics and time between first and second assessment for each group.

	2D (*n* = 21)	3D (*n* = 20)	AR (*n* = 20)	Total (*n* = 61)
Mean age (SD)	41.1 (14.6)	36.7 (15.7)	41.1 (15.1)	39.6 (15.1)
Gender				
• Men (%)	7 (33)	5 (25)	9 (45)	21 (34)
• Women (%)	14 (67)	15 (75)	11 (55)	40 (66)
Level of education				
• Lower vocational training (%)	1 (5)			1 (2)
• General secondary education (%)		2 (10)	2 (10)	4 (7)
• Higher professional training (%)	13 (62)	10 (50)	10 (50)	33 (54)
• Academic degree (%)	7 (33)	8 (40)	8 (40)	23 (37)
Pre-existing anatomical knowledge				
• 0 (none)	4 (19)	6 (30)	7 (35)	17 (28)
• 1 (television)	11 (52)	6 (30)	5 (25)	22 (36)
• 2 (books, own line of work)	5 (24)	7 (35)	7 (35)	19 (31)
• 3 (doctor)^*^	1 (5)	1 (5)	1 (5)	3 (5)
Professional background				
• Healthcare	6	5	7	18
• Education	6	8	3	17
• Sales, hospitality, recreation	2	3	2	7
• Construction	2	0	1	3
• Information technology (IT)	0	1	2	3
• Other/student	5	3	5	13
Mean amount of days between first and second assessment (SD)	8.8 (1.7)	8.2 (1.5)	9.1 (2.6)	8.7 (2.0)

There was no difference in age, level of education or pre-existing anatomical knowledge between groups. 2D = two-dimensional group (MRI), 3D = three-dimensional group (3D model), AR = Augmented Reality group. * = other than neurologist/neurosurgeon.

**Table 2 jcm-09-03660-t002:** Outcome of questionnaires and ninety-five percent confidence interval.

	2D (*n* = 21)	3D (*n* = 20)	AR (*n* = 20)	Total (*n* = 61)
Immediate recall (95% CI)	83.60 (81.74–85.47)	89.91 (88.52–91.30)	87.97 (86.14–89.81)	87.10 (85.95–88.26)
1-Week Follow-up (95% CI)	83.05 (80.02–86.08)	88.06 (86.41–89.72)	84.89 (82.02–87.79)	85.30 (83.78–86.81)

There was an overall higher score at immediate recall (*p* = 0.001). Both 3D (*p* < 0.0005) and AR (*p* = 0.025) group scored significantly higher than the 2D group. 2D = two-dimensional group (MRI), 3D = three-dimensional group (3D model), AR = Augmented Reality group.

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
