# Peer review of "Information Recall in Pre-Operative Consultation for Glioma Surgery Using Actual Size Three-Dimensional Models"

_jcm, 2020, doi:10.3390/jcm9113660_

Round 1

Reviewer 1 Report

In this study, Sezer et al present an experimental study which compares the amount of information recalled after a mock pre-operative consultation for glioma surgery of healthy individuals. The authors compare a conventional 2D Model using MRI and fMRI images with a printed and a virtual 3D Models. The authors conclude that the information recalled improves when using both 3D models in comparison to the 2D model.

I would like to congratulate the authors for presenting an interesting and well constructed study for this relevant topic. The manuscript is well written and interesting to read.

Here are my remarks:

Page 5, line 140: I suppose MRI group corresponds to 2D group. Please use uniform labeling for the groups.

As the authors mention in the introduction, 40-80% of information provided to the patient is immediately forgotten. In their study, the recall is well over 80% in all study groups even one week after consultation (table 2), which is in my opinion not terribly bad for a 2D model. Do the authors consider it to be due to the participation of healthy individuals or well-structured consultation?

In line 207-208, the authors recommend the use of a physical (3D printed) model as superior due to the larger effect sizes. Nevertheless, the production of a 3D model is time and resource consuming, compromising its practicality in everyday clinical setting. In my opinion, the authors should mention that in the discussion.

Reviewer 2 Report

This is a well thought out, studied and presented paper. It could benefit from more referencing to the legal implications / benefits of improving recall pre-operatively. I also think that many centres would not be able to afford to print the 3D models, especially with the pending world recession due to COVID. The costs of printing a 3D model and cost of developing the institution's own 3D VR modelling software for an iPad may be worth discussing in the context of their cost-effectiveness

Syntax errors:

Page 1 line 17 - groups, not group;  line 18 - participants (not participant)

Page 1 line 22 - 'have the ability to' may be better write 'appear to'

Consider adding numbers of participants in the abstract

Page 2 line 67 - consider 'level of a professional neurologist, neurosurgeon or neuroscientist'

Page 3 line 191 - statistically, not statistic
